



.

# Porting the WAVEWATCH III (v6.07) Wave Action Source Terms to GPU

Olawale James Ikuyajolu[1,2], Luke Van Roekel[3], Steven R Brus[4], Erin E Thomas[3], and Yi Deng[1,2]

[1]Earth and Atmospheric Sciences, Georgia Institute of Technology, Atlanta, GA, USA
[2]Program in Ocean Science and Engineering, Georgia Institute of Technology, Atlanta, Georgia, USA
[3]Fluid Dynamics and Solid Mechanics (T-3), Los Alamos National Laboratory, Los Alamos, NM, USA
[4]Mathematics and Computer Science Division, Argonne National Laboratory, Lemont, IL, USA

**Correspondence:** Olawale James Ikuyajolu (oikuyajolu3@gatech.edu)




**Abstract.** Surface gravity waves play a critical role in several processes, including mixing, coastal inundation and surface fluxes. Despite the growing literature on the importance of ocean surface waves, wind-wave processes have traditionally been excluded from Earth system models due to the high computational costs of running spectral wave models. The Next Generation Ocean Model Development in the DOE's (Department of Energy) E3SM (Energy Exascale Earth System Model) project partly focuses on the inclusion of a wave model, WAVEWATCH III (WW3), into the E3SM. WW3, which was originally developed for operational wave forecasting, needs to be computationally less expensive before it can be integrated into ESMs. To accomplish this, we take advantage of heterogeneous architectures at DOE leadership computing facilities and the increasing computing power of general-purpose graphics processing units (GPU). This paper identifies the wave action source terms as the most computationally intensive module in WW3 and then accelerates them via GPU. Using one GPU, our experiments on two computing platforms, Kodiak (P100 GPU & Intel(R) Xeon(R) CPU E5-2695 v4) and Summit (V100 GPU & IBM POWER9), show speedups of up to 2.4x and 6.6x respectively over one MPI task on CPU. Using different combinations of multiple CPUs and GPUs, we obtained an average speedup of 2x and 4x on Kodiak and Summit. We also discuss how the trade off between occupancy, register and latency affects the GPU performance of WW3.





# 1 Introduction

Ocean surface gravity waves, which derive energy and momentum from steady winds blowing over the surface of the ocean (Hasselmann, 1991), are a very crucial aspect of the physical processes at the atmosphere-ocean interface. They influence a variety of physical processes such as momentum and energy fluxes, gas fluxes, upper ocean mixing, sea spray production, ice fracture in the marginal ice zone and Earth albedo (Cavaleri et al., 2012). Such complex wave processes can only be treated accurately by including a wave model into Earth system models (ESMs). The first ocean models neglected the existence of ocean waves by assuming that ocean surface is rigid to momentum and buoyancy fluxes from the atmospheric boundary layer (Bryan and Cox, 1967). Currently, most state-of-the-science ESMs are still missing some or all of these wave-induced effects (Qiao et al., 2013) despite the growing literature on their in the simulation of weather and climate.

Recent literature has shown that incorporating different aspects of surface waves into ESMs lead to improved skill performance, particularly in the simulation of sea surface temperature, wind speed at 10m height, ocean heat content, mixed layer depth, and the Walker and Hadley circulations (Law Chune and Aouf, 2018; Song et al., 2012; Shimura et al., 2017; Qiao et al., 2013; Fan and Griffies, 2014; Li et al., 2016). Yet, only two climate models that participated in the Climate Model Intercomparison Project phase 6 (CMIP6), i.e. the First Institute of Oceanography-Earth System Model version 2 (FIO-ESM v2.0; Bao et al., 2020) and the Community Earth System Model Version 2 (CESM2; Danabasoglu et al., 2020) have a wave model as part of their default model components. However for CMIP6, only FIO-ESM v2.0 employed a wave model. Wind-wave induced physical processes have traditionally been excluded from ESMs due to the high computational cost of running spectral wave models on global model grids for long term climate integrations. In addition to higher computing costs due to longer simulation times, adding new model components also increases resource requirements e.g. number of CPUs/nodes. The Next Generation Ocean Model Development in the US DOE's (Department of Energy) E3SM (Energy Exascale Earth System Model) project, partly focuses on the inclusion of a spectral wave model, WAVEWATCH III (WW3), into the E3SM to improve the simulation of coastal processes within E3SM. To make WW3 within E3SM feasible for long term global integrations, we need to make it computationally less expensive.

Computer architectures are evolving rapidly, especially in the high-performance computing environment, from traditional homogeneous machines with multicore central processing units (CPUs) to heterogeneous machines with multi-node accelerators such as graphics processing unit (GPUs) and multicore CPUs. Moreover, the number of CPU-GPU heterogeneous machines in top 10 of the TOP500 supercomputer increases from 2 in November 2015 (https://www.top500.org/lists/top500/2015/11/) to 7 in November 2021 (https://www.top500.org/lists/top500/2020/11/). The advent of heterogeneous super-computing platforms, together with the increasing computing power and low energy to performance ratio of GPUs, has motivated the use of GPUs to accelerate climate and weather models. In recent years, numerous studies have reported successful GPU porting of full or partial weather and climate models with improved performances (Hanappe et al., 2011; Xu et al., 2015; Yuan et al., 2020; Zhang et al., 2020; Bieringer



et al., 2021; Mielikainen et al., 2011; Michalakes and Vachharajani, 2008; Shimokawabe et al., 2010; Govett et al., 2017; Li and Van Roekel, 2021; Xiao et al., 2013; Norman et al., 2017, Norman et al., 2022; Bertagna et al., 2020). GPU programming model is totally different than CPU code, so programmers must recode or use directives to port codes to GPU. Because climate and weather models consist of million lines of code, a majority of the GPU-based climate simulations only operate on certain hot spots (most computationally operations) of the model while

leaving a large portion of the model on CPUs. In recent years, however, efforts have been made to run an entire model component on the GPU. For example, Xu et al. (2015) port the entire Princeton Ocean Model to GPU and achieved a 6.9x speedup. Similarly, the entire E3SM atmosphere including SCREAM (Simple Cloud-Resolving E3SM Atmosphere Model) is running on the GPU (https://github.com/E3SM-Project/scream). Taking advantage of the recent advancements of GPU programming in climate sciences together with the heterogeneous architectures

at DOE leadership computing facilities, this study seeks to identify and move the computationally intensive parts of WW3 to GPU through the use of OpenACC pragmas.

    The rest of the paper is structured as follows. In Section 2, we first present an overview of the WW3 model and its parallelization techniques, give an introduction to the OpenACC programming model, describe the hardware and software environment of our testing platforms, and finally present the test case configuration used in this

study. The result section, Section 3, presents WW3 profiling analysis on CPU, discusses the challenges encountered, GPU-specific optimization techniques and compares the GPU results with the original FORTRAN code. Section 4 concludes the paper.

## 2   Model Description and Porting Methodology

### 2.1   WAVEWATCH III

WW3 is a third-generation spectral wave model developed at the US National Centers for Environmental Prediction (NOAA/NCEP) (Komen et al., 1994) from the WAve Model (WAM) (The Wamdi Group, 1988). It has been used widely to simulate ocean waves in many oceanic regions for various science and engineering applications (Chawla et al., 2013b; Alves et al., 2014; Cornett, 2008; Wang and Oey, 2008). To propagate waves, WW3 solves the random phase spectral action density balance equation, $N(\phi, \lambda, \sigma, \theta, t)$, for wavenumber-direction spectra. The intrinsic fre-

quency ($\sigma$) relates the action density spectrum to the energy spectrum (F), $N = \frac{F}{\sigma}$. For large scale application, the evolution of the wave action density in WW3 is expressed in spherical coordinates as follows:

$$\frac{\partial N}{\partial t} + \frac{\partial (C_\phi N)}{\partial \phi} + \frac{\partial (C_\lambda N)}{\partial \lambda} \frac{\partial (C_\sigma N)}{\partial \sigma} + \frac{\partial (C_\theta N)}{\partial \theta} = \sum_i S_i \qquad (1)$$

    Where $F$ is the Energy density; $\phi$ is the longitude; $\lambda$ is the latitude; $\sigma$ is the relative frequency; $\theta$ is the direction; $t$ is the time and $S$ represent the source and sinks terms. The net source-sink terms consist of several physical

processes responsible for generation, dissipation and redistribution of energy. The net source-sink terms available in





WW3 are waves generation due to wind ($S_{in}$), dissipation ($S_{ds}$), non-linear quadruplet interactions ($S_{nl}$), bottom friction ($S_{bt}$), and depth-limited breaking ($S_{db}$), Triad wave-wave interactions ($S_{tr}$), scattering of waves by bottom features ($S_{sc}$), wave-ice interactions ($S_{ice}$), reflection off shorelines or floating objects($S_{ref}$), and a general purpose slot for additional, user defined source terms ($S_{user}$). Details of each source term can be found in the WW3 manual
(WAVEWATCH III® Development Group, 2019). The primary source-sink terms used in this work are $S_{in}$, $S_{ds}$, $S_{nl}$, $S_{bt}$ and $S_{db}$. Several modules are used for the calculation of source terms. However, module `w3srcemd.ftn` manages the general calculation and integration source terms. Eqn. 1 is solved by discretizing in both physical space ($\lambda$, $\phi$) and spectral space ($\sigma$,$\theta$).

When moving code between different architectures, it is necessary to understand its structure. For the purposes of
our study, Fig. 1 shows a representation of the WW3 algorithm structure. WW3 is divided into several submodules, but the actual wave model is the `w3wavemd`, which runs the wave model for a given time interval. Within `w3wavemd`, several modules are called at each time interval to handle initializations, interpolation of winds and currents, spatial propagation, intra-spectral propagation, calculation and integration of source terms, output file processing, etc. In our work we found that `w3srcemd` is the most computationally intensive part of WW3, thus we focus our attention
on this module. According to Fig. 1a, `w3srcemd` is being called at each spatial grid point, which implies that the spatial grid loop is not contained in `w3srcemd`, but rather in `w3wavemd`. Fig. 1b, which represents `w3srcemd`, contains a dynamic integration time loop that can only be executed sequentially. It also calls a number of submodules, such as `w3src4md` for computation of the wind input and wave breaking dissipation source terms. `w3srcemd` consists of collapsed spectral loops ($NSPECH = NK \times NTH$), where NK is the number of frequencies ($\sigma$) and NTH is the number of
wave directions ($\theta$). Lastly, the `w3src4md` Fig. 1c consists of only frequency (NK) loops. The structure of other source terms submodules is similar to `w3src4md`.

## 2.2  WW3 Grids and Parallel Concepts

The current version of WW3 can be run and compiled for both single and multi-processor (MPI) compute environments with regular grid, two-way nested (mosaic) grids (Tolman, 2008), spherical multi-cell (SMC) grids (Li, 2012),
and unstructured triangular meshes (Roland, 2008, Brus et al., 2021). In this study, we ran and compiled WW3 using MPI with unstructured triangular meshes as the grid configuration. In WW3, the unstructured grid can be parallelized in physical space using either Card Deck (CD) (Tolman, 2002) or domain decomposition (Abdolali et al., 2020). Following Brus et al. (2021), we used the CD approach as the parallelization strategy. The ocean (active) grid cells are sorted and distributed linearly between processors in a round-robin fashion using $n = \mod(m-1, N)$ i.e.
grid m is assigned to processor n. Where N is the total number of processor and M is the total number of ocean grids. If N is divisible by M, every processor n has the same number of grids, NSEAL (Fig. 1a). The source term calculation as well as the intra-spectral propagation are computed using the aforementioned parallel strategy, but data are gathered on a single processor to perform the spatial propagation.



## 2.3 OpenACC

To demonstrate the promise of GPU computing for WW3, we used the OpenACC programming model. OpenACC
is a directive-based parallel programming model developed for engineers and scientists to run codes on accelerators
without significant programming effort. Programmers incorporate compiler directives in the form of comments into
FORTRAN, C, or C++ source codes to assign the computationally intensive the sections of the code to be executed
on the accelerator. OpenACC helps to simplify GPU programming because the programmer is not preoccupied

with the code parallelism details, unlike CUDA and OpenCL where you need to change the code structure to
achieve GPU compatibility. OpenACC compiler automatically transfer calculations and data between two different
architectures, the host (CPU) and the accelerator device (GPU). OpenACC works together with OpenMP, MPI and
CUDA, supporting heterogeneous parallel environments. Starting from version 4.5, the OpenMP API (Application
Programming Interface) specification has been extended to include GPU offloading and GPU parallel directives.

While OpenMP and OpenACC have similar constructs, OpenMP is more prescriptive. Prescriptive directives describe
the exact computation that should be performed and provide the compiler no flexibility. In the long run, it is unclear
whether vendors other than NVIDIA will support OpenACC. Our study focuses solely on OpenACC since it is the
most mature API for NVIDIA GPUs at the time of analysis.

OpenACC has three-levels of parallelism (Fig. 2) namely: vector, worker and gangs, corresponding to thread, warp

and block in CUDA terminology . A gang is group of workers, where multiple gangs can work independently without
synchronization. Workers are groups of vector/threads within a gang and vector is the finest level of parallelism
operating with single-instruction, multiple data (SIMD). Gang, worker, and vector can be added to a loop region
needed to be executed on GPU. An example of Fortran code with and without OpenACC directives is shown in
Fig. 3. The OpenACC directives are shown in green as comments starting with !$acc (e.g. lines 6 & 12). In Fig. 3a,

line 6 is a declaration directive for allocating memory for variables on GPU, line 8 is the data region to move data
into the GPU, line 11 updates data already present on GPU with new values from the CPU, line 12 launches the
parallel region, line 8 updates CPU data with new values from the GPU and line 28 deletes the data on GPU after
computation. To learn more about all OpenACC directives, please refer to the NVIDIA (2017) or Chandrasekaran
and Juckeland (2017)

## 2.4   Test Case Configuration


In this study, the WW3 model was configured and simulated over the global ocean with an unstructured mesh of
of 1°global resolution and 0.25°in regions with depth less than 4km e.g. 1°at the equator and 0.25°at the coastal
regions (Brus et al., 2021). The number of the unstructured mesh nodes is 59,072 (hereafter 59K). In addition, we
demonstrate the effect of scaling the problem size on speedup by using an unstructured mesh with 228,540 nodes

(hereafter 228K). For both spatial grid configurations, the spectral grid has 36 directions and 50 frequency bands
that range exponentially from 0.04 to 0.5 $Hz$, separated by a factor of 1.1 (Chawla et al., 2013a). In WW3, the



combinations and types of the source terms in Eqn. 1 depends on the research question being answered. WW3 has several source term packages which can be implemented by activating different switches. However, since the goal of this study is purely computational, we selected the commonly used source-sink terms switches, ST4, DB1, BT1

and NL1. The ST4 switch (Ardhuin et al., 2010) consist of the wind input ($S_{in}$) and wave breaking dissipation ($S_{ds}$) source terms, DB1 switch is for the depth-induced breaking ($S_{db}$) source term, BT1 switch consist of bottom friction ($S_{bt}$) source term parameterizations and the non-linear quadruplet wave interactions ($S_{nl}$) are computed in the model using the NL1 switch. For the detailed description of each source term switch in WW3, the reader is referred to the WW3 user manual (WAVEWATCH III® Development Group, 2019).

To accelerate WW3 on GPU, we employed two computational platforms with heterogeneous architectures, namely:

1. Kodiak cluster from the Parallel Reconfigurable Observational Environment (PROBE) (Gibson et al., 2013) of Los Alamos National Laboratory. Kodiak has 133 compute nodes. Each node contains Intel(R) Xeon(R) CPU E5-2695 v4 @ 2.10GHz 36 CPU cores and 4 NVIDIA Tesla P100 SXM2 GPGPUs, each with 16 GB of memory.

2. Summit, a high performance computing system at the Oak Ridge National Laboratory at Oak Ridge National Laboratory (ORNL). Summit has 4,608 nodes, each contains 2 IBM POWER9 CPUs and 6 NVIDIA Tesla V100 GPUs, each with 80 streaming multiprocessors. All connected together with NVIDIA's high-speed NVLink. Summit is the fastest supercomputer in the US and the second fastest in the world.

Table 1 shows the configuration of each compute node for both platforms. For a more accurate comparison of
CPU and GPU codes, we used the same compiler. On Kodiak, the CPU FORTRAN code was compiled using the flags `-g -O3 -acc`. Similarly, the OpenACC code was compiled with flags `-g -O3 -acc -Minfo=accel -ta=tesla,ptxinfo,` `maxregcount:n`. Likewise on Summit, the flags for the CPU code are `-g -O2`, and that of the OpenACC are `-g -O2 -acc` `-ta=tesla,ptxinfo,maxregcount:n -Minfo=accel`. Options `-ta=tesla:ptxinfo,maxregcount:n` are the optimization flags used in this study (Section 3.2). Adding the option `-ta=tesla:ptxinfo` to the compile flags provides information about the
amount of shared memory used per kernel (a function that is called by the CPU for execution on the GPU) as well the registers per thread. The flag `-ta=tesla:maxregcount:n`, where "n" is the number of registers, sets the maximum number of registers to use per thread.

As a test case, we performed a 5-hour simulation from 2005/06/01 00:00:00 - 05:00:00 by forcing WW3 with atmospheric winds derived from the US National Center for Atmospheric Research renalysis (NCAR). We validate
the GPU model using simulated significant wave heights (SWH).





## 3    Results and Evaluations

In this section, we first describe the performance of WW3 on CPU and its computationally intensive sections. Furthermore, we discuss the challenges encountered and how GPU optimization is done. Lastly, we present the performance result of porting WW3 on GPU.

### 3.1    WW3 Profiling Analysis on CPU

With model optimization, an important step is to find the runtime bottlenecks by measuring the performance of various sections in units of time and operations. In order not to waste time and resources improving the performances of rarely used subroutines, we first need to figure out where WW3 spends most of its time. The technical term for this process is called profiling. For this purpose, we profile WW3 by running the Callgrind profiler from the Valgrind tool and then visualize the output using a KCachegrind tool. An application's performance can be 10 to 50 times slower when profiling it with callgrind, however, the proportions of times remain the same. Figure 4 shows the callgraph obtained by profiling WW3 with 8 MPI processes. The source term subroutine, W3SRCEMD, can easily be spotted as the consumer of 78% of the total execution time and resources. Within the W3SRCEMD subroutine, the dissipation source term $S_{ds}$ uses more than 40% of the total runtime because it contains numerous time consuming spectral loops. In fact, profiling WW3 with another profiler (not shown), Intel Advisor (Intel Corporation , 2021), specifically highlights the spectral time consuming loops. In WW3, each processor serially run through its sets of allocated spatial grids (described in Section 2.2) with each containing a spectral grid points, and `w3srcemd` has a time-dynamic integration (Fig. 1b) procedure which can not be parallelized. Looping through spectral grids and the time-dynamic integration procedure are plausible reasons why `w3srcemd` is the bottleneck of WW3. We name WW3 model with GPU accelerated W3SRCEMD as WW3-W3SRCEMD.gpu and WW3.cpu as the CPU only version.

Fortunately, W3SRCEMD does not contain neighboring grid dependencies in both spatial and spectral i.e. no parallel data transfers, and is therefore suitable for GPU porting. We moved the entire WW3 source terms computation to GPU as shown in Fig. 3b.

### 3.2    Challenges & Optimization

Once the program hotspot is parallelized and ported to the GPU, the GPU code needs to be optimized in order to improve its performance. Conventionally, optimization of GPU codes involves loop optimization (fusion and collapse), data transfer management (CPU to GPU and GPU to CPU), memory management and occupancy. Some of these optimization techniques are interrelated e.g. memory management and occupancy. The WW3 model contains very few collapsible loops, so loop fusing and loop collapsing did not effectively optimize the code (figure not shown). To successfully port WW3-W3SRCEMD.gpu and achieve the best performance, two challenges had to be overcome in this study. The first is a data transfer issue caused by the WW3 data structure, while the second is a memory management and occupancy issue caused by the use of massive arrays to store spatial-spectral data sets.





### 3.2.1  Data Transfer Management

It is important to understand the layout of data structures in the program before porting to GPU. WW3 outlines
its data structures by using modules e.g. `w3adatmd, w3gdatmd, & w3odatmd` (lines 2 & 3 of Fig. 3). Depending on the
variable required, each subroutine uses these modules. These variables are called global external variables. Since
it is not possible to move data within a compute kernel in GPU, all necessary data must be present on the GPU
prior to launching the kernel that calls W3SRCEMD. The structure of WW3 requires the use of routine directive
(`!$acc routine`) to create a device version of `W3SRCEMD`, as well as other subroutines in it. In FORTRAN, the routine
directive appears in the subprogram specification section or its interface block. Due to the use of routine directives,
OpenACC declare directives(`!$acc declare create`; line 6 of Fig. 3b) were added to the data structures module to
inform the compiler that global variables need to be created in the device memory as well as the host memory.

All data must be allocated on the host before being created on a device, unless it has been declared device
resident. In WW3, however, arrays whose size are determined by spatial-spectral grid information are allocated at
runtime rather than within the data structure modules. Thus, creating data on the device within the WW3 data
modules poses a problem. Due to this restriction, it was then necessary to explicitly move all the required data to
GPU before launching the kernel. Therefore, we could not assess the impact of implicit data transfers on the GPU
performance. However, copying data at each time step or at the start of time integration typically consumes less
than 0.1% of GPU time. For managing data transfers between iteration cycles, we use `!$acc update device(variables)`
and `!$acc update self(variables)` (lines 11 & 21 in Fig. 3b). However, it would be easier to use unified memory, which
offers a unified memory address space to both CPUs and GPUs, rather than tracking each and every data that needs
to be sent to the GPU, but the latest OpenACC version does not support the use of unified memory with routine
directives. In the future, having this feature would save time spent tracking data transfers in programs with many
variables such as WW3.

### 3.2.2  Memory management and Occupancy

Occupancy is defined as the ratio of active warps (workers) on a streaming multiprocessor (SM) to the maximum
number of active warps supported by the SM. On Kodiak and Summit, the maximum threads per SM is 2048. A
warp consists of 32 threads which implies that the total number of possible warps per SM is 8. Even if the kernel
launch configuration maximizes the number of threads per SM, other GPU resources, such as shared memory and
registers, may also limit the number of maximum threads, thus indirectly affecting GPU occupancy. A register is a
small amount of fast storage available to each thread, whereas shared memory is the memory shared by all threads
in each SM. As seen in Table 1, the maximum memory per SM for Kodiak and Summit is 256KB, but the maximum
shared memory for Kodiak is 64KB (P100) and 94KB for Summit (V100). In terms of total register file size per
GPU, Kodiak and Summit have 14336KB and 20480KB, respectively.






In this study, the kernel launch configurations consist of NSEAL gangs, where NSEAL is the number of grids on each node, and 32 vector lengths, implying that each gang consists of 32 threads. With this configuration, the achievable GPU occupancy (regardless of other resources) is 50%. To estimate GPU usage, we use the CUDA occupancy calculator available on https://docs.nvidia.com/cuda/cuda-occupancy-calculator/CUDA_Occupancy_Calculator.xls.

Adding `-ta=tesla:pxtinfo` to the compiler flags gives the information about the size of registers, memory spills (movement of variables out of the register space to the main memory) and shared memory allocated during compilation. Kodiak and Summit both allocate the maximum 255KB register per thread, reducing GPU occupancy to 13%. In addition, a full register leads to spilling of memory into the L1 cache (shared memory). A spill to cache is fine, but a spill to the global memory will severely affect performance because the time required to get data from the global

memory is longer than that from a register, latency. Latency is the amount of time required to move data from one point to the other. Therefore, an increase (decrease) in register size causes two different things simultaneously: a decrease (increase) in occupancy and decrease (increase) in latency. There is always a trade off between register, latency and occupancy, and the the trick is to find the spot that maximizes performance. One can set the maximum number of registers per thread via the flag `-ta=tesla:maxregcount:n` where "n" is the number of registers. Figure 8

illustrates how the trade off between latency and occupancy affects the GPU performance based on the number of registers. Our analysis was based only on the performance of running 228K grid configuration with 16 MPI tasks. For Summit, as the number of registers increases from 16 to 64, the GPU occupancy remains constant at 50% and GPU performance improved due to the movement of more variables to the fast memory. As indicated by the blue part of line, this is a latency-dominant region. However, as the number of register increases from 64 to 192, GPU

occupancy gradually decreases from 50% to 13%. This degraded performance despite moving more data into the fast memory. Therefore, this is an occupancy-dominant region as indicated by the red part of the line (Fig. 8b). From 192 to the maximum register count, the occupancy remains constant at 13% and GPU performance remains relatively constant. As occupancy is constant, latency is expected to dominate this region, which is represented by the green part of the line in Fig. 8b. However, we observe no memory spill for registers between 192 and 255 and thus

latency effect remains unchanged. Therefore, constant latency and occupancy leads to the constant performances in this region.

Fig. 8b shows that 64 registers produced the best performance (minimum runtime) on Summit. For brevity, the previously described trade-off analysis also applies to Kodiak, and 96 register count produced the best performance (fig. 8a). On Kodiak, the register count that achieved the best performance is higher than on Summit, probably due

to Summit's larger L1 and L2 caches (Table 1). With larger L1 and L2 caches, more data can be stored, reducing memory spillover to global memory and thus reducing latency. Optimizing the register count increases the GPU performance by approximately 20% on Kodiak and 14% on Summit.





### 3.3 GPU Accelerated W3SRCEMD

This section compares the performance of WW3.cpu and WW3-W3SRCEMD.gpu. In this study, one CPU and one GPU correspond to one MPI task each. For each platform and grid configuration used in this study, Figure 5 shows the execution time in seconds and performance improvement between the GPU-accelerated and CPU-only versions. Comparing the performance of a single GPU with a single CPU on Kodiak, a 2.3x & 2.4x speedup was achieved for 59K (Fig. 5a) and 228K (Fig. 5b) grids respectively. Similarly on Summit, we achieved speedups of 4.7x & 6.6x for 59K (Fig. 5c) and 228K (Fig. 5d) grids respectively. On Summit, the GPU performance of 228K nodes is better because the CPU gets extremely slow. Summit's speedup is greater than Kodiak's because the Tesla V100-SXM2-16GB GPU is faster than the Tesla P100-PCIE-16GB GPU (NVIDIA, 2017).

We also measure the GPU's strong scaling performance by fixing the grid size and doubling the number of CPUs/GPUs. Strong scaling is a measure of how performance changes as a function of the number of processors for a given problem size. For the 228K nodes, Fig. 6 shows the scaling efficiency of WW3 on CPU and GPU. On Kodiak (Summit), the scaling efficiency is 93.8% (98.3%) on two CPUs, 90.5% (98.4%) on four CPUs and 83.5% (142%) on eight CPUs. While it increases above 100% for 16 CPUs and above on Summit, it decreases to 72% for 64 CPUs on Kodiak. The CPU scaling efficiency is consistent with the parallel efficiency behaviour described in Fig. 3 of Tolman (2002). On GPU, the strong scaling efficiency in Kodiak (Summit) is 86% (97%) on two GPUs, 81% (95%) on four GPUs and 83% (94%) on eight GPUs. The scaling efficiency drops rapidly as the number of GPUs increases, e.g. 64 GPUs have 67% (72%) on Kodiak (Summit). CPUs and GPUs exhibit the same scaling efficiency pattern as their numbers increase. Remember that in strong scaling, the number of processors is increased while the problem size remains the same. As a result, it reduces the workload (distributed grids) per CPU/GPU. Due to a reduced workload, GPU utilization falls, so the performance advantage over CPU decreases gradually (Fig. 5). Larger numbers of GPUs or CPUs seem to yield little improvement in run time.

Note that the speedups in this section are achieved by parallelizing the local grids loop, which calls the W3SRCEMD function, using the OpenACC parallel directives (Fig. 3b). In `W3SRCEMD` and its dependent subroutines, we introduced the OpenACC routine directive (`!$acc routine`) which instructs the compiler to build a device version of the subroutine so that it may be called from a device region by each gang. In addition, at the start of the time integration, we moved the needed constants data to the GPU (line 8 of Fig. 3b). Using the average over the late 2-hour simulation, Fig. 7 compares the output results of CPU and WW3-W3SRCEMD.gpu codes and their relative difference for significant wave height (SWH). According to the validation results, the SWH is nearly identical, and the error is negligible and acceptable. It is possible that the error stems from the difference in mathematical precision between the GPU and CPU.





## 4  Conclusions

305  Climate science is increasingly moving towards higher spatial resolution models and additional components to better simulate previously paramaterized or excluded processes. In recent decades, the use of GPUs to accelerate scientific problems has increased exponentially due to the emergence of supercomputers with heterogeneous architectures.

Wind generated waves play an important role in modifying physical processes at the atmosphere ocean interface. They have generally been excluded from most coupled Earth system models partly due to its high computational 310  cost. However, the Energy Exascale Earth System Model (E3SM) project seeks to include a wave model (WW3) and introducing WW3 to E3SM would increase the computational time and usage of resources.

In this study, we identified and accelerated the computationally intensive section of WW3 on GPU using OpenACC. Using Valgrind & callgrind tools, we found that the source term subroutine, W3SRCEMD, consumes 78% 315  of the execution time. The W3SRCEMD subroutine has no neighboring grid points dependencies, and is therefore well suited for implementation on GPU. On two different computational platforms, Kodiak with a P100 GPU and Summit with a V100 GPU, we performed 5-hour simulation experiments using two global unstructured meshes with 59,072 and 228,540 nodes. Our results showed an overall GPU speedup of 2.4x (2.3x) and 6.6x (4.7x) relative to serial CPU on Summit and Kodiak for 228,540 (59,072) nodes respectively. On average, running W3RCE on GPU 320  gives approximately 4x (2x) performance gain, in terms of both time and resources, over CPU version on Summit (Kodiak). For example on Kodiak (Summit), the execution time for 16 GPUs is approximately equals to that with 32 (64) CPU (MPI tasks). Therefore, leveraging heterogeneous architectures reduces the amount of time and resources required to include WW3 in E3SM. However, the biggest limitation to GPU performance gain is the trade off between registers and latency. Too many constants in WW3 occupy the register (fast memory) and then spill 325  over to the L1 and L2 caches or the GPU global memory.

Typically, CMIP pre-industrial control experiments are run for  500 years often with more than 20 ensemble members. According to our results, this corresponds to a potential speedup of several days of walltime. In the future, we plan to examine the performance of GPU-accelerated WW3 using OpenMP and estimate the amount of effort 330  it will take to convert our OpenACC parallelization structure into OpenMP. The success of this work has laid the foundation for future work in global spectral wave modeling, and it is also a major step toward expanding E3SM's capability to run with waves on heterogeneous architectures in the near future.

*Code and data availability.* Model configuration and input files can be assessed at https://doi.org/10.5281/zenodo.6483480 .The official repository of WAVEWATCH III CPU code can be found here: https://github.com/NOAA-EMC/WW3. The new 335  WW3-W3SRCEMD.gpu code used in this work is available at https://doi.org/10.5281/zenodo.6483401



*Author contributions.* OJI was responsible for the code modifications, simulations, verification tests and performance analyses. LVR developed the concept for this study. OJI provided the initial draft of the manuscript. LVR, SRB, EET and YD contributed to the final manuscripts

*Competing interests.* The authors declare that they have no conflict of interest.

340   *Acknowledgements.* This research was supported as part of the Energy Exascale Earth System Model (E3SM) project, funded by the U.S. Department of Energy, Office of Science, Office of Biological and Environmental Research. This research used resources provided by the Los Alamos National Laboratory Institutional Computing Program, which is supported by the US Department of Energy National Nuclear Security Administration under contract no. 89233218CNA000001. This research used resources of the Oak Ridge Leadership Computing Facility at the Oak Ridge National Laboratory, which is supported by the
345   Office of Science of the U.S. Department of Energy under Contract No. DE-AC05-00OR22725.



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





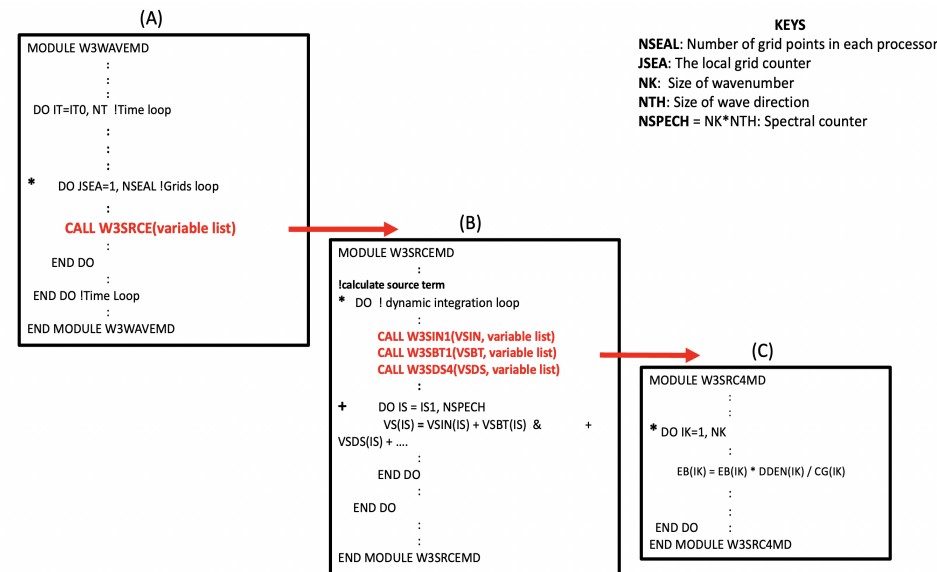

**Figure 1.** A schematic representation of WAVEWATCH III code structure





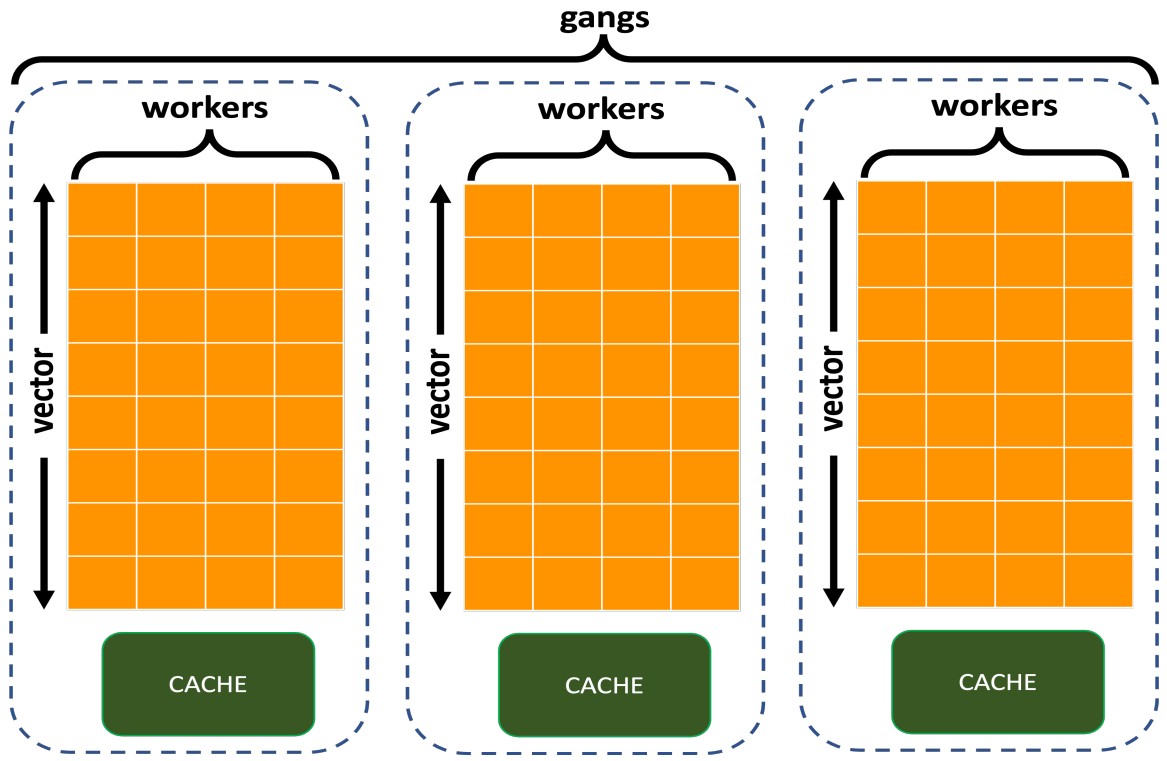

**Figure 2.** The map of gangs, workers, and vectors (adapted from Jiang et al., 2019)





```fortran
 MODULE W3WAVEMD
 USE CONSTANTS
 USE W3GDATMD
 !! use other modules !!
 REAL       :: Variables
 !! ============= !!
 !! BUNCH OF CODES !!
 !! ============= !!
DO IT=IT0, NT !Time loop
!! ============= !!
!! BUNCH OF CODES !!
!! ============= !!
DO JSEA=1, NSEAL !grids loop
!! BUNCH OF CODES !!
CALL W3SRCEMD(variable list)
!! BUNCH OF CODES !!
END DO
!! ============= !!
!! BUNCH OF CODES !!
!! ============= !!
END DO !Time Loop
!! ============= !!
!! BUNCH OF CODES !!
!! ============= !!
END MODULE W3WAVEMD
```

(a)

```fortran
 MODULE W3WAVEMD
 USE CONSTANTS
 USE W3GDATMD
 !! use other modules !!
 REAL       :: Variables
 !$acc declare create(needed variables on GPU)
 !! BUNCH OF CODES !!
 !$acc enter data copyin(variables to GPU)
 DO IT=IT0, NT !Time loop
!! BUNCH OF CODES !!
!$acc update device(GPU data)
!$acc parallel copy(NSEAL) num_gangs(NSEAL)
!$acc loop gang
DO JSEA=1, NSEAL !grids loop
!! BUNCH OF CODES !!
CALL W3SRCEMD(variable list)
!! BUNCH OF CODES !!
END DO
!$acc end parallel
!acc update host(needed variables on CPU)
!! ============= !!
!! BUNCH OF CODES !!
!! ============= !!
END DO !Time Loop
!! BUNCH OF CODES !!
!$acc exit data delete(all GPU variables)
END MODULE W3WAVEMD
```

(b)

**Figure 3.** A schematic representation of WAVEWATCH III Original FORTRAN source code for the W3WAVEMD module (a) and its OpenACC directives version (b)



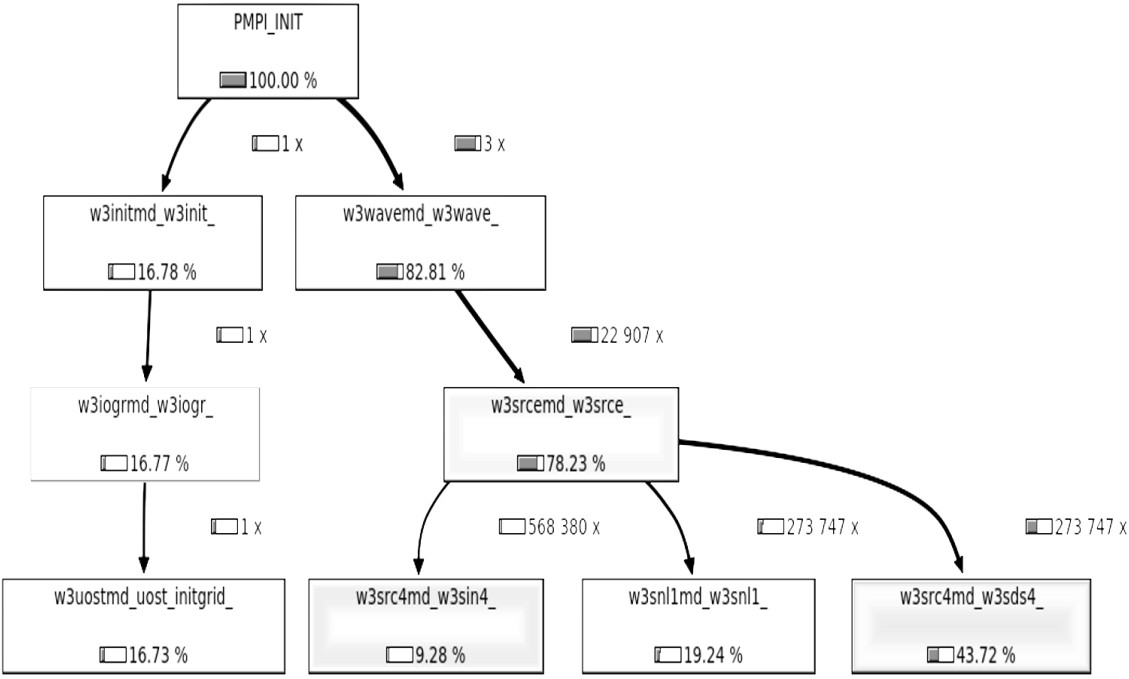

**Figure 4.** The Callgraph obtained by profiling WAVEWATCHIII with 8 CPUs. Each box includes the name of the subroutine and its relative execution time as a percentage





**Figure 5.** The runtime of WW3.cpu (orange) and WW3-W3SRCEMD.gpu (green) for different numbers of GPU and CPU on Kodiak with 59072 (a) and 228540 (b) grids, and on Summit with 59072 (c) and 228540 (d) grids. The black line represents the speedups of WW3-W3SRCEMD.gpu over WW3.cpu. one CPU and one GPU correspond to one MPI task each.



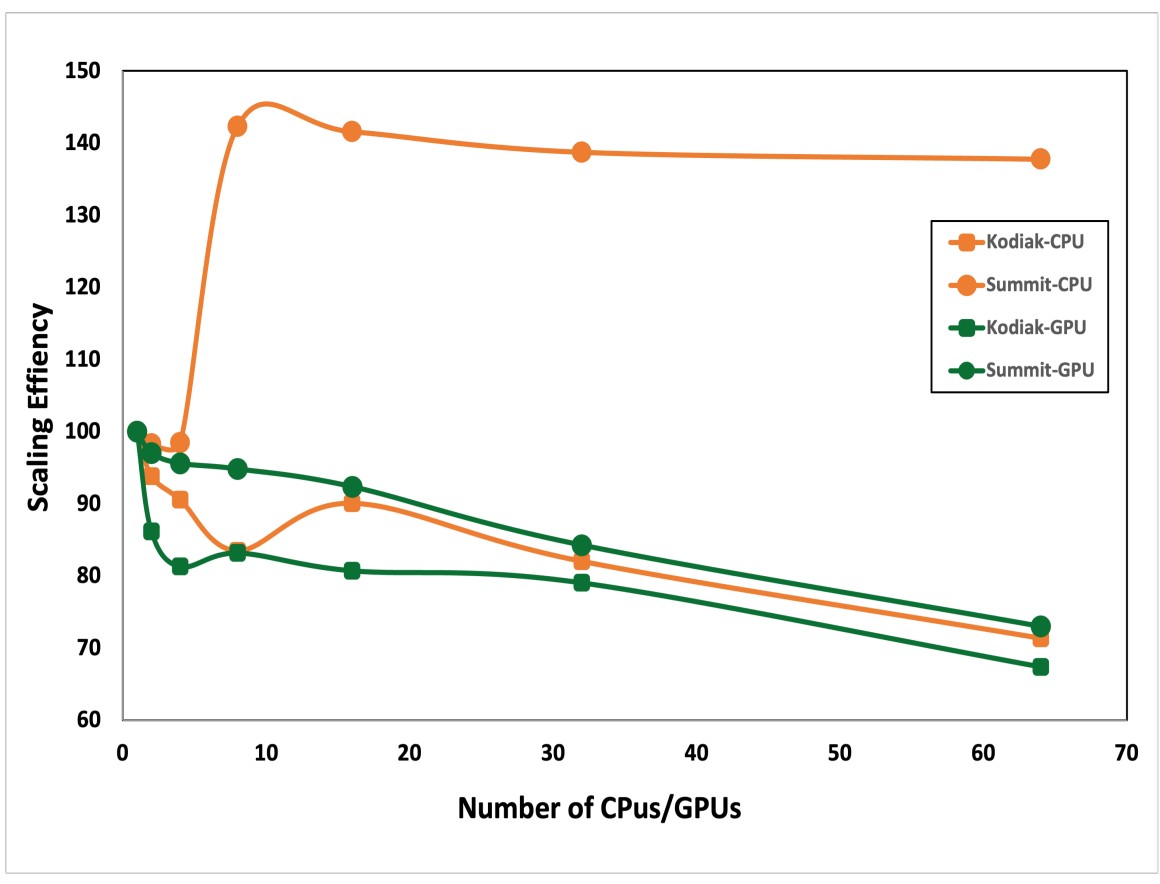

**Figure 6.** The strong scaling result of WW3-W3SRCEMD.gpu and WW3.cpu on Kodiak and Summit

**Figure 7.** The average of WW3.cpu (top), WW3-W3SRCEMD.gpu (middle) last 2-hour simulations and their differences (bottom) for significant wave height

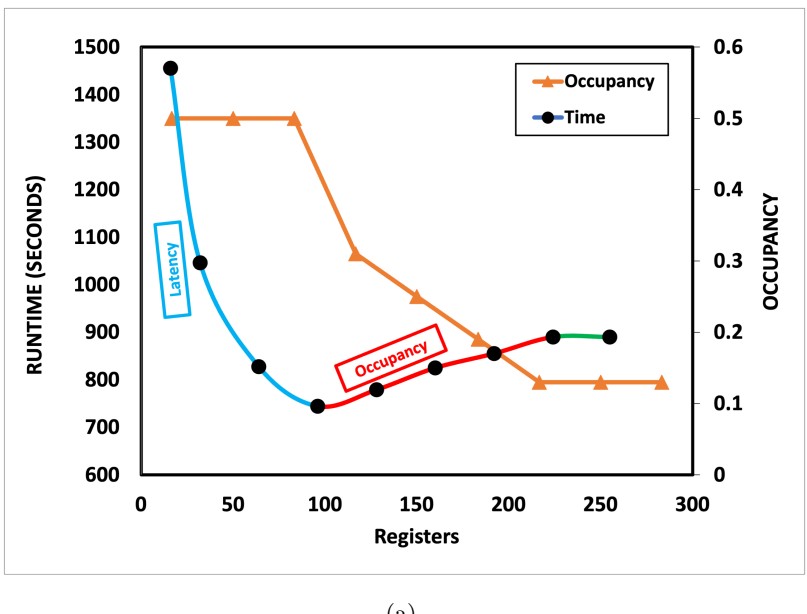

(a)

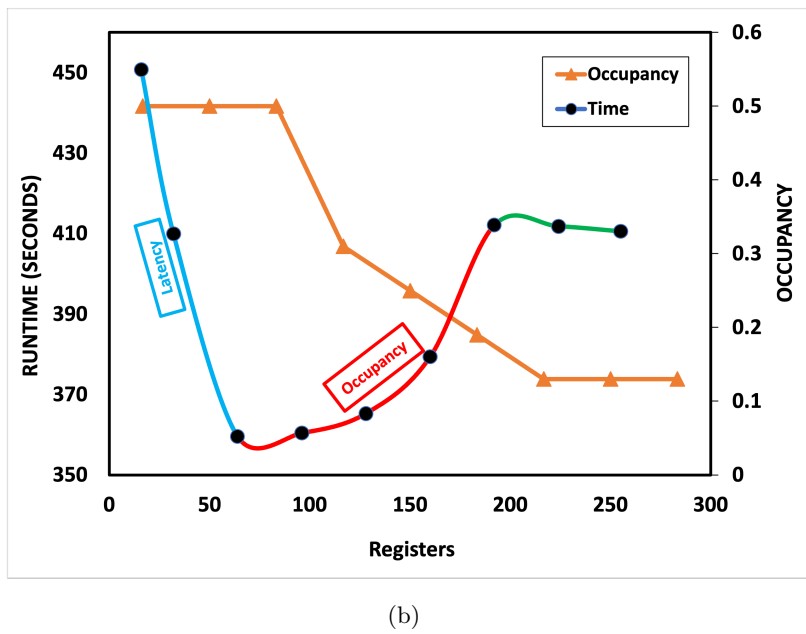

(b)

**Figure 8.** WW3-W3SRCEMD.gpu runtime (primary vertical axis) and occupancy (secondary vertical axis) based on register counts on Kodiak (a) and Summit (b). For the runtime, the light blue part of the line represents the latency-dominant region, the red part represents the occupancy-dominant region, and the green part represents the neutral region. We analyzed the result of running 228540 grids on 16 GPUs and CPUs.



**Table 1.** GPU hardware specifications.

|  | SUMMIT (V100) | KODIAK (P100) |
| --- | --- | --- |
| Compute Capability | 7 | 6 |
| Global memory size | 16 GB | 16 GB |
| L1 cache | 10 MB | 1.3 MB |
| L2 cache | 6 MB | 4 MB |
| Shared memory size / SM | Configurable up to 96 KB | 49 KB |
| Constant memory | 64 KB | 64 KB |
| Register File Size | 256 KB (per SM) | 256 KB (per SM) |
| 32-bit Registers | 65536 (per SM) | 65536 (per SM) |
| Max registers per thread | 255 | 255 |
| Number of multiprocessors (SMs) | 80 | 56 |
| Warp size | 32 threads | 32 threads |
| Maximum resident warps per SM | 64 | 64 |
| Maximum resident blocks per SM | 32 | 32 |
| Maximum resident threads per SM | 2048 | 2048 |
| Maximum threads per block | 1024 | 1024 |