# Peer review of "Porting the WAVEWATCH III (v6.07) Wave Action Source Terms to GPU"

_Geoscientific Model Development, 2022_

## Author Comment (AC1)

**Reply to Reviewer 1**

*General Remarks:*

This manuscript presents results from a modified WAVEWATCH III code, which off-loads the spectral source terms to GPUs. The authors investigate the scaling and two different platforms, and validates the results against a CPU-only run.

This is a very timely, important and well written manuscript. I can recommend that it is accepted for publication after some minor changes. I also want to mention, that the way the problem is broken down and presented step by step made the manuscript easy to follow. Please find my comments and questions below:

We wish to thank the reviewer for their positive comments regarding the manuscript, particularly for the explicit acknowledgment of the writing structure. We also thank the reviewer for the insightful questions and suggestions raised. Detailed answers have been provided to the questions and revision has been done following the suggestions. Below are the original comments and the changes we have made in the manuscript to respond to the comments. Also, we have added a node-level baseline/comparison (using all the CPU cores and GPUs on a node) section to the manuscript. The baseline comparison was done only on the summit machine because KODIAK LANL has been decommissioned.

*Specific Comments:*

lines 24-25: "despite the growing literature on their in the simulation of weather and climate."

Seems to be missing a word

Yes, we have updated the manuscript to include the missing word. "despite the growing literature on their **importance** in the simulation of weather and climate."

line 71: I'm not sure what this is a reference to, but usually Komen et al. 1994 is used as a WAM reference?

We have corrected the reference. (WAVEWATCH III® Development Group, 2019)

line 86-88: These sentences are slightly confusing, since first we are talking about modules that calculate source terms (right hand of Eq. 1), and then we talk about discretizing (left hand side).

The sentence has been reorganized, and the discussion of discretization has been moved immediately following Eqn (1)

line 94: Can the relative computational intensivness change if we are using defferent propagation schemes. Can extremely small time steps alter/tip this balance? (Probably not, I guess.)

As shown in Fig. 1 (updated Fig. 4 in the manuscript), the source terms approximately take 82% out of 96% of the W3WAVEMD computational time. This means that the remaining computations in

W3WAVEMD (propagation scheme inclusive) ~14%. Thus, different propagation schemes won't alter the balance that much.

Moreover, with the unstructured grid configurations, we are limited to the lowest order scheme (CRD N-Scheme) because it is well-tested and more robust for realistic applications (Abdolali et al., 2020; Roland, 2008). The source term time step is dynamically adjusted based on local wave properties, but when running on a global domain, the effects of the time step dependencies are averaged out.

[Figure]

Figure 1: The Callgraph obtained by profiling WAVEWATCH III with 300 MPI ranks. Each box includes the name of the subroutine and its relative execution time as a percentage

line 188-189: I'm a bit surprised that Sin takes half of the source term computational, time, since I would have expected the non-linear interactions to be the heaviest. The cumulative breaking term in ST4 is supposed to be quite resource consuming, while possible not having a large effect on the end results. Is that turned on or off here? If it's turned on, then that would perhaps be an obvious candidate to try to speed up the model (not directly related to GPU porting).

Going back to ST3 would definitely speedup the code on both CPU and GPU. However, it is possible that the GPU gain some speedup over CPU due to the reduction of divergence among threads heavily present in ST4 code. We only focus on ST4 term since it is widely used for global wave modeling.

line 196-198 Is this a hard requirement, or does the lack of communication just mean that the source terms are trivially paralellisizable? The word "suitable" suggests that any communication here would make GPU porting a non-option, but I'm not sure that is the case (altough it probably becomes a lot more complex).

Yes, you are right. It would only make the porting more complex. We have changed the sentence to '**Fortunately, W3SRCEMD does not contain neighboring grid dependencies in both spatial**

**and spectral i.e. no parallel data transfers, W3SRCEMD can therefore be ported to GPUs with less difficulty**.' – Line 202 -205

line 241-266: very long paragraph. Even though the paper is generally very well written, this aspect could be checked.

We have modified it according to the suggestion. The paragraph has been split into two.

line 275: It seems like the order the figures are presented might be wrong, since Fig. 8 has already been referenced?

We have rearranged the figures.

line 295: Would it be possible to increase occupancy by reorganizing the loops? Now we loop over all grid points, and then loop over one spectrum, but perhaps it would be more efficient to define an array that has both spatial and spectral dimensions (and perhaps slice that up into some blocks, if needed)? Can you comment on this?

Although it is possible, it would require a significant amount of code refactoring, and that would defeat the whole purpose of using OpenACC in the first place, which is to avoid excessive refactoring. However, moving forward, this is necessary to achieve more speedups.

The paper is missing a discussion section. Although it might not strictly be needed in this kind of more technical paper, it would perhaps be interesting for the reader to know what kind of impact these speed-ups might have in practical terms. Several days of wall time was mentioned, but is this a "game changer" to allow for including wave models in ESMs, or do we still need to optimize? I'm also wondering how well the exact non-linear solution might scale (if the authors can comment), since this might have consequences to very basic reasearch into e.g. wave growth that might be affected by the crude approximations of DIA. Finally, would it every be viable to port any other parts of the wave model, such as the propagation, to GPUs, or is the communication needed beween the grid points a complete deal breaker?

Currently, on the CPU, WW3 in E3SM is computationally intensive. As E3SM is expanding other model components' capabilities to run on heterogeneous architecture, and depending on the speed-up achieved for other components, it might be necessary to further optimize WW3 code.
While there is still room for further GPU optimizations, such as pushing down the grid loop counter into the W3SRCEMD, this would require major code refactoring. There are other parts of WW3 code that can be ported to GPU, such as spatial propagation, and GPU communication is not a deal breaker, but it might require more bookkeeping. According to Fig. 1 (Fig. 4 in the manuscript), a significant speed-up should not be expected.

**References**

Abdolali, A., Roland, A., Van Der Westhuysen, A., Meixner, J., Chawla, A., Hesser, T. J., Smith, J. M., and Sikiric, M. D.: Large-scale hurricane modeling using domain decomposition parallelization and implicit scheme implemented in WAVEWATCH III wave model, Coast. Eng., 157, 103656, https://doi.org/10.1016/j.coastaleng.2020.103656, 2020

Roland, A. Development of WWM II: Spectral wave modelling on unstructured meshes. Diss. Ph. D. thesis, Technische Universität Darmstadt, Institute of Hydraulic and Water Resources Engineering, 2008.

---

## Author Comment (AC2)

**Reply to Reviewer 2**

**General comments**

The paper is well formulated in that the problem of interest is well described, the approach to GPU porting is well described, and some key aspects of performance are explained well in detail. I am requesting major revisions to the manuscript due to the importance of using appropriate baselines for GPU performance reporting. If the GPU performance is much slower than most CPUs, as seems to be the case, it is a very important data point for the reader to understand clearly.

Comparing GPU runtime against a single CPU core is inappropriate because in production simulations, the entire CPU would have been used. I request that the authors change this comparison to use all available CPU cores in the CPU baselines when reporting speed-up numbers.

We thank the reviewer for pointing out the need for a better baseline comparison to understand GPU performance. We have completed a detailed node-level performance comparison of GPU and CPU resources on the Summit supercomputer and included in Section 3.3.1. Using all 42 CPU cores and 6 GPUs on a summit node (with 7 MPI ranks per GPU), a speedup of 1.4x (1.36x) is achieved for a mesh size of 228K (59K). We also ran 4, 3 and 2 MPI ranks per GPU configurations. The speedups of all the GPU configurations are compared with the full CPU 42 cores run (with 7 MPI ranks per GPU). The results (Table 3 in revised manuscript) of the different multi-process configurations are included in the revised manuscript. The revised manuscript also includes an updated Figure 6.

| GRID | CPU | GPU | | | |
|---|---|---|---|---|---|
| | 42 ranks | 42 ranks (7/GPU) | 24 ranks (4/GPU) | 18 ranks (3/GPU) | 12 ranks (2/GPU) |
| 59K | 180.72 | 133.18 [1.36x] | 124.28 [1.45x] | 127.17 [1.42x] | 146.34 [1.23x] |
| 228K | 683.66 | 483.22 [1.41x] | 523.91 [1.30x] | 533.08 [1.24x] | 612.60 [1.12x] |
| 228K (W3SRCEMD) | 589.56 | 389.12 [1.52x] | 407.54 [1.45x] | 366.00 [1.61x] | 395.31 [1.49x] |

Table 3. Node baseline comparison runtime (seconds) with different MPI ranks per GPU configuration on a Summit node. The bold numbers in [ ] are the speedup relative to the whole CPU on a summit node (42 MPI ranks). In the last row, we present the speedups and simulation times when comparing only the GPU-accelerated subroutine, W3SRCEMD.

On the same note, I request that the authors include a "roofline" plot of their ported kernel. This can be obtained directly using Nvidia's ncu-ui tool (or several other tools if desired). Absent this, then the authors need to at least provide the floating point operations per second (flop/s) achieved by the

kernel as well as the maximum expected flop/s for the observed DRAM-oriented arithmetic intensity in their kernel. This is a more objective performance metric because the baseline is fixed for a given kernel and GPU hardware choice. The documents below should help in doing this:
https://www.nersc.gov/assets/Uploads/Talk-NERSC-NVIDIA-FaceToFace-2019.pdf
https://arxiv.org/pdf/2009.02449.pdf

We wish to thank the reviewer for their positive comments and suggestions raised regarding the baseline comparison.

From the roofline model (Fig. 8 in the revised manuscript), the kernel is limited by the data transfer bandwidth between the CPU and the GPU. The kernel has a very low arithmetic intensity for both simple and double-precision floating-point (Fig. 8) computations, performing only very few flops for every double and integer loaded from DRAM. Most of the kernel's time is spent in executing memory (load/store) instructions. This is due to too many local scalars and arrays in the W3SRCEMD subroutine and the huge WW3 memory requirement. For example, VA, the spectra storage array, is approximately 5Gb (20Gb) for a spatial grid size of 59,000 (228,000) and spectral resolution of 50x36. VA is one of the large arrays used in W3SRCEMD. W3SRCEMD is simply too big and complicated to be ported using OpenACC routine directives and therefore requires significant code refactoring. We have modified the manuscript based on the node comparison and roofline analysis. In sections 3.3.1 and 3.4 of the revised manuscript, we provide a detailed explanation of the baseline comparison and roofline analysis, respectively.

[Figure]

Figure 8. Roofline model for W3SRCEMD kernel from Nvidia Nsight Compute. The red and green dots are the double and single precision values respectively.

*Specific comments*

Line 52: The phrase "totally different" is not an accurate statement. The "breadth-first" SIMT dispatch of code over threads on GPUs is different than the more "depth-first" execution of code of CPUs. GPUs need a larger degree of parallelism exposed at one time than CPUs. Beyond this, much of the programming and optimization approach does remain the same. The order of execution as dispatched on the hardware should still match the order of memory accesses in arrays. Floating point and integer divisions and floating-point transcendental operators should be minimized. Data movement to and from DRAM should be minimized.

Here, we are referring to just the structure of the code. CPU codes need to be modified by adding directives and in some cases require little or more refactoring. We have removed the 'totally' to avoid confusion.

Line 127: It might be more accurate to say that OpenACC contains the most mature implementation using the Nvidia compiler suite on Nvidia GPUs.

We have modified the sentence according to the reviewer's suggestion.

Line 129: These lists do not correspond to one another in a respective sense; and I believe, as written, this will lead to confusion for the reader. An unspoken yet commonly used mapping from OpenACC levels of parallelism to CUDA levels of parallelism is as follows: gang == blockIdx.x (i.e., "grid"-level parallelism); worker == threadIdx.y (i.e., "block" level parallelism); and vector == threadId.x (i.e., finer "block"-level parallelism). Perhaps a better more general statement is something similar to "Gang, worker, and vector parallelism expose increasingly fine granularities of parallelism to distribute work over grid, block, warp, and thread-level parallelism on Nvidia GPUs." That should be true in all cases for the OpenACC spec itself and Nvidia hardware.

We have modified the sentence according to the reviewer's suggestion.

Line 130: Gangs must operate independently without synchronization.

We have removed the 'can' in the sentence to emphasize this.

Line 132: SIMD is not an accurate description of the parallel dispatch strategy on Nvidia GPUs. Please describe it as "SIMT" (single instruction, multiple thread).

We have modified the word according to the reviewer's suggestion.

Line 135: Please specify that declare create is needed specifically due to using module-level variables directly in device code instead of passing them by parameter.

We already stated this in section 3.2.1.

Line 138: Please add that the expectation is that "W3SRCEMD" will then further dispatch parallel threads in the worker and/or vector levels.

We have modified the sentence.

Line 167: Can you give the reasoning for reducing the optimization on Summit? If bugs were encountered, this can be useful information for the reader to understand that these issues can crop up sometimes.

Yes, the -O3 optimization flags do not run for summit. Worked with the OLCF support but couldn't figure why. It does not affect the conclusion. We have added this as a footnote to the revised manuscript.

Line 207: Occupancy is largely affected by register and shared memory usage. Using large local, thread-private arrays (i.e., on the stack) can affect register usage, but I believe it is incorrect to say that occupancy is affected by the size of the module-level arrays created outside the kernel. Can the authors explain in more detail what is meant here?

The sentence should be 'Many local arrays, sometimes of spectral length, and scalars used within W3SCRE and its embedded subroutines'. We have modified this in the revised manuscript.

Line 222: I may be misunderstanding this, but it's not clear why the impact of data transfers could not be assessed. Nvidia's nvvp and nsight tools should be able to show the cost of all transfers.

The sentence has been removed since we have determined that CPU-GPU data transfers dominate the program.

Line 242: Using only 32 threads per gang seems like it would lead to problems hiding memory fetching latency from DRAM via thread switching within an SM. Have you tried increasing this to 64 or 128, or is 32 something required by the algorithm itself?

Yes, 32 is something required by the algorithm probably due to register or shared memory usage.

Line 252: I think it's important to note at this point that there is another option that has, so far, not been discussed. The NSEAL loop could be pushed down the callstack into the innermost routines. While this would be a significant refactoring effort, it would allow developers to fission the one large kernel into multiple smaller kernels. This will increase the number of kernel launches (potentially increasing an important kernel launch latency cost), but each individual kernel would no longer suffer register spillage, which is likely the number one performance problem in the approach used in this paper. I believe this approach should at least be mentioned in the manuscript so that the reader understands there are multiple potential approaches.

Yes, there is room for further GPU optimization by pushing down the grid loop counter into the W3SRCEMD which would require major code refactoring. Moreover, this is the only way to reduce register usage, increase GPU occupancy and avoid launching a single large W3SRCEMD kernel. We have included this potential in the conclusion section of the manuscript.

Line 257: There is not only a maximum number of registers per thread, but I believe there is a minimum number supported by the hardware as well, which may explain this behavior.

Figure 8: The plot seems confusing because a blue line is represented for "time" in the legend, which make it seem like the red and green portions are potentially no longer representing "time". Perhaps add two entries to the legend so that the labels are "Occupancy", "Time (latency regime)", "Time (occupancy regime)", and "Time (neutral regime)".

The figure (Fig. 5) has been modified based on the reviewer comments.

[Figure]

(a)

[Figure]

(b)

Figure 5: WW3-W3SRCEMD.gpu runtime (primary vertical axis) and occupancy (secondary vertical axis in orange) based on register counts on Kodiak (a) and Summit (b). For the runtime, the light blue part of the line represents the latency-dominant region, the red part represents the occupancydominant region, and the green part represents the neutral region. We analyzed the result of running 228K mesh size

Line 274: I do not consider this to be an appropriate performance comparison. A GPU should not be compared to a single CPU core. It should be compared to an entire CPU with reasonable optimization efforts performed on both the CPU and GPU. For instance, if the GPU code is 6-7x faster than a single P9 core, then when using the P9 as it would typically be used (21-42 cores), the V100 performance is actually 3-6x slower than a single P9.

Once again, we would like to thank the reviewer for bringing this up. We have added a node level baseline/comparison section to the manuscript. The baseline comparison was done only on the summit machine because KODIAK LANL has been decommissioned. Section 3.3.1

Line 300: I greatly appreciate considerations of correctness in this paper. This is often overlooked in GPU refactoring manuscripts.

We would like to thank the reviewer for the kind words

Line 307: "exponentially" is likely an inaccurate term. Perhaps use "significantly" or a similar word instead

We have modified the sentence.

Line 318: Please mention the baseline here. Line 320 seems to indicate that the comparison is against an entire CPU whereas the manuscript seems to indicate that it is against only one core of the CPU.

We have modified the conclusion and manuscript to focus on the baseline comparison.

Line 323: The authors should mention briefly here the other potential approach described in the comment for line 252 above.
Line 324: Are the authors certain the register usage is due largely to constants? What is the evidence for this claim?

W3SRCEMD subroutine has several large subroutines (based on the type of source term switches selected) and they make use of many scalars and constants.

---

## Author Comment (AC3)

**Reply to Reviewer 3**

**General Comments:**

The authors present a GPU acceleration of the source term part within the parallel context of the WW3 Framework. The work is innovative and important. However, there are some flaws within the theoretical approach and the tests that have been done when evaluating the new parallelization option.

In eq. (1) the wave action equation we have certain terms that are local parts (e.g. source terms and spectral advection), and global parts, which is the geographical advection. The geo. advection needs some parallel exchange either for the CD or the DD approach.

Now, when expanding this asymptotically using Amdahl's law for the given problem it can easily be seen that ultimatievly for a infinite number of computational cores the only cost that remains would be the parallell exchange since all other workload tends to zero.

Introducing now the communicators to the GPU, this would remain as well an overhead and add up to the global exchange of the advection part itself.

Now the most important question is how does the scheme scale for various constellation. Since the GPU layer was introduced the scalability analysis becomes twodimensional in terms of number of GPU and CPU. This question remain open in this paper even if the authors have sufficient acces to the needed computational resources. In the sense of the above also the quantification of the computation cost of the source terms is rather linked to give testcase constellation investigated in this paper.

I conclusion I think that the work is interesting and the implemenation is a important topic for GMD but the work lacks in depth scalability analyis and therefore no final statment can be made in terms of efficiency. Especially, in the context that only 8 cores have been used from possible NSPEC cores within the CD approach.

In conclusion I think that much more work must be done to evaluate the performance of this approach before the given conclusion can be made and the general contribution of the work can be evaluated, which is now not the case.

We wish to thank the reviewer for their comments and suggestions regarding the manuscript. The focus of this study is to accelerate WW3 for global modeling and prepare the wave model for the E3SM exascale regime. In light of this, we focused on the current WW3 configurations in E3SM and only at the most computationally intensive region, which is the source term (Fig. 1). This is true for any propagation schemes or grid parallelization concepts at least when using ST4 source term. Other parts of WW3 can benefit from GPU, but as the first GPU work in spectral wave modeling, our aim is to focus on the expensive region first. As suggested by the reviewer, we have also included a scaling analysis up to 5 nodes (i.e. 210 CPU cores and 30 GPUs) in the manuscript. Table 4 in the manuscript shows the results of scaling the 228K mesh size over multiple nodes. We used the full GPU configuration here by launching 7 MPI ranks on each GPU. In the results, it can be seen that the speedup is relatively uniform across multiple nodes.

[Figure]

Figure 1: The Callgraph obtained by profiling WAVEWATCH III with 300 MPI ranks. Each box includes the name of the subroutine and its relative execution time as a percentage

Specific Comments:

1. 94: "is the most computationally intensive part of WW3" Can this be quantified? Moreover, I can not agree on this, since this would be rather linked to the given configuration and the used schemes. For implicit schemes and various other constellations with high resolution geographical space this must not be true. I think that too much general statements have been derived by the given configuration and testcase. I could imagine that with high spatial resolution and a lot of computational cores, which have unfortunately not been used here, the communication itself will take more time than the computation of the source terms itself, as explained in the general comments above.

The authors are using CD for their parallelization strategy, but I do not see why one could not use domain decomposition in combination with GPU. How would the scheme perform with DD approach? Why so few computation cores. Why so much development and such little evaluation of the performance?

The authors have limited their simulation to just 8 processors, however, I do not see why it could not be used on say 1000 processors if each of those processors has access to a GPU. This cannot be repeated often enough. It remains the major concern of this work. The Kodiak and Summit supercomputer have several thousand processors. Why is the limit to 8?

The interaction between explicit and implicit computations is not considered. I would think that the code can be used for explicit and implicit and the scalability should be evaluated for both.

Further code moving to the GPU could be the frequency shifting and refraction in explicit mode, has this been considered as well?

We focused on the current WW3 grid parallelization method in E3SM. The number of processors used in profiling does not change the fact that the source term is the most expensive region. Figure 1 shows the callgraph obtained by profiling WW3 with 300 MPI ranks. The source term is still the most computationally intensive region. We have updated Fig. 4 in the revised manuscript. Since this is the first attempt to run WW3 GPU, future studies would focus more on how different WW3 setup (grid parallelization method, source term switches, propagation schemes, e.t.c.) impacts GPU speedup over the CPU.

We have conducted sensitivity studies (performed on only CPUs, at this point) to determine the potential speed up due to domain decomposition (as opposed to card deck decomposition) parallelization methods as well as the impact of explicit computations (as opposed to implicit). We found domain decomposition and implicit computations perform significantly worse (fewer Simulated Years Per Day (SYPD)) than explicit, card deck computations for the global E3SM grid used in this study (results shown in Figures 2 and 3 below). We believe this is due to the fact that the E3SM mesh considered here is smaller and has a less drastic difference between the minimum and maximum resolution compared to the mesh used in Abdolali et al. 2020.

[Figure]

Figure 2. Simulated Years Per Day (SYPD) of Explicit, Domain Decomposition (blue line) and Explicit, Card Deck Decomposition (orange line) simulations of WW3 plotted as a function of the number of tasks. For direct comparison, the simulations are plotted by the number of tasks normalized by the spectral resolution. Note that the number of tasks used by Explicit, Card Deck Decomposition is limited by the spectral resolution (the maximum tasks used cannot exceed the spectral resolution). Explicit, Card Deck Decomposition clearly outperforms Explicit, Domain Decomposition.

[Figure]

Figure 3. Simulated Years Per Day (SYPD) of Explicit, Card Deck Decomposition (blue point), and Explicit, Domain Decomposition (black point) and Implicit, Domain Decomposition as a function of time step (red line). The explicit simulations use a time step of 900 seconds, while the time step of the implicit simulations varies between 900 seconds and 2600 seconds. Increasing the time step of the implicit simulations, increases the SYPD of WW3 until about 1800 seconds, beyond which, the time step does not speed up (or increase the SYPD) of WW3.

Technical issues:

- There is a typo in Equation (1), the "+" is missing

  We have corrected the typo.

  The literature reference in the paper is not done properly. Here are two examples:

- 19: Either referer to the paper with „e.g." or put the original reference, which is not Hasselmann 1991. It was Gelci et al. 1957 if i remember right.
- 146: Again, Chawla et al. Was referenced but those authors did not derive the scaling of 1.1. The original publication should be cited Hasselmann & Hasselmann, 1981 or it can be add „e.g.". However, the latter I do not find very useful since we want to honor and cite the original work. This should be cleaned throughout the paper.

  We are attributing our entire spectral mesh settings to Chawla and not the 1.1 factor. We have modified the sentence

Missing reference:

- 104: Brus et al., 2021 reference is missing.

  We have corrected the missing reference (Line 111)

All references have been checked.